# VLM-Grounder: A VLM Agent for Zero-Shot 3D Visual Grounding

**Runsen Xu**[1,3]   **Zhiwei Huang**[2]   **Tai Wang**[3]   **Yilun Chen**[3]   **Jiangmiao Pang**[3✉]   **Dahua Lin**[1,3,4]

[1]The Chinese University of Hong Kong   [2]Zhejiang University   [3]Shanghai AI Laboratory
[4]Centre for Perceptual and Interactive Intelligence

**Abstract:** 3D visual grounding is crucial for robots, requiring integration of natural language and 3D scene understanding. Traditional methods depending on supervised learning with 3D point clouds are limited by scarce datasets. Recently zero-shot methods leveraging LLMs have been proposed to address the data issue. While effective, these methods only use object-centric information, limiting their ability to handle complex queries. In this work, we present VLM-Grounder, a novel framework using vision-language models (VLMs) for zero-shot 3D visual grounding based solely on 2D images. VLM-Grounder dynamically stitches image sequences, employs a grounding and feedback scheme to find the target object, and uses a multi-view ensemble projection to accurately estimate 3D bounding boxes. Experiments on ScanRefer and Nr3D datasets show VLM-Grounder outperforms previous zero-shot methods, achieving 51.6% Acc@0.25 on ScanRefer and 48.0% Acc on Nr3D, without relying on 3D geometry or object priors. Codes are available at https://github.com/OpenRobotLab/VLM-Grounder.

**Keywords:** 3D Visual Grounding, VLM Agent, Zero-Shot Scene Understanding

## 1   Introduction

3D visual grounding focuses on finding the 3D location of a target object in a scene based on user queries, which is a fundamental requirement for robots. This task requires integrating natural language understanding with 3D scene comprehension. Previous methods mainly rely on supervised learning using paired 3D point clouds and language data to train end-to-end models. However, existing visual grounding datasets[1, 2] are scarce and limited to a pre-defined vocabulary, challenging the development of general models for open-world applications.

To address this issue, recent approaches [3, 4] have utilized large language models (LLMs) [5, 6, 7, 8, 9] in a zero-shot manner for 3D visual grounding. Since LLMs cannot directly process 3D environments, these methods employ a point cloud-based 3D localization module [10, 11] to detect objects and convert their attributes into texts. The LLM then selects the target object based on these texts, as illustrated in Fig. 1. While these methods achieve strong performance, they use only object-centric information and often miss detailed scene context, making it challenging to handle queries like "find the room with the most abundant natural light."

Inspired by the recent advancements in vision-language models (VLMs) [12, 13, 14, 15, 16] that excel in directly associating language with visual information, we introduce **VLM-Grounder**, an agent framework based on VLMs for zero-shot 3D visual grounding. Our approach involves a VLM that analyzes user queries and sequences of images capturing the scene to locate the target object, whose 2D mask is projected to determine the 3D bounding box.

Inputting image sequences to the VLM can exceed the VLM's maximum image limit, overly consume the VLM's context length, and lead to degraded performance and increased inference latency. Stitching multiple images is an effective solution, but it may result in information loss. We design a novel Visual-Retrieval benchmark to quantitatively evaluate how different stitching layouts affect the

8th Conference on Robot Learning (CoRL 2024), Munich, Germany.

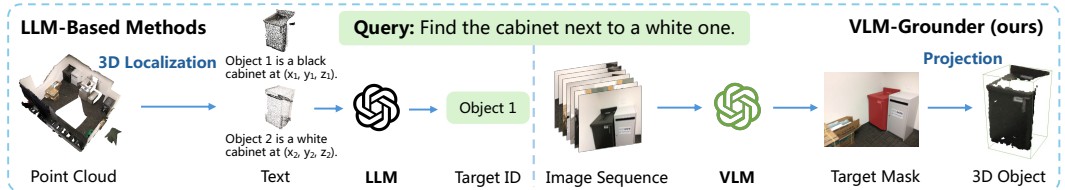

Figure 1: **Comparison between LLM-based methods and VLM-Grounder.**

VLM's visual processing. Further, we propose a dynamic stitching strategy that dynamically uses the optimal layouts identified by the benchmark to stitch images, enhancing VLM's performance.

Given user queries and stitched images, the VLM is responsible for finding the target object. To fully utilize the VLM's reasoning capabilities, we develop a grounding and feedback scheme. In this scheme, the VLM explains its reasoning process across image sequences. Automatic feedback is provided to for retrying when the VLM gives an invalid response, ensuring more accurate outcomes.

After the target object is found, we extract the fine-grained 2D mask and get the 3D bounding box by projection. However, estimating a 3D bounding box from a single image can be problematic due to limited field-of-view and inaccurate depth information. We design a multi-view ensemble projection module that uses image matching to find additional views of the same target object. These views are then used together to jointly estimate the 3D bounding box. Additionally, we employ morphological operations to better handle issues related to inaccurate depth.

We conducted extensive experiments on the widely used ScanRefer [1] and Nr3D [2] datasets. Our VLM-Grounder outperforms previous zero-shot methods and is even comparable with some supervised learning methods, without relying on 3D geometry, such as point clouds or provided object priors. Specifically, VLM-Grounder achieves an overall Acc@0.25 of 51.6% on the ScanRefer benchmark and 48.0% overall accuracy on the Nr3D benchmark, surpassing the previous SOTA methods' performances of 36.4% and 39.0%, respectively.

## 2 Related Work

**3D visual grounding.** 3D visual grounding was first benchmarked by ScanRefer [1] and ReferIt3D [2] based on ScanNet [17] static point clouds, requiring the output of the target object's 3D bounding box specified by a language description. Previous supervised-learning methods primarily follow a two-stage paradigm: a 3D detection model [1, 18, 19, 20, 21, 22, 23] or a 3D instance segmentation model [24, 25, 26] generates object proposals, and a language branch encodes the user query for feature fusion with object features to predict target objects. There are one-stage methods[27, 28] directly decoding the target object bounding box by an encoder-decoder architecture. Recently, large language models (LLMs)[9, 7, 8, 29, 30, 31, 5, 32] have been employed as backbones for selecting or decoding target objects[33, 34, 35, 36]. Unlike end-to-end models, zero-shot methods leverage LLMs in an agent-based framework. LLM-Grounder [4] parses user queries to identify the target object type and referenced object type, then uses an open-vocabulary semantic segmentation model [10, 11] to locate these object types. An LLM is then used to reason which object satisfies the grounding relationship. ZS3DVG [3] follows a similar pipeline but requires the LLM to write codes to determine the target object. Despite their great performance, these methods rely on reconstructed point clouds and are bottlenecked by 3D localization modules. Additionally, they only process text-based information with the LLM and primarily address user queries involving spatial relationships.

**Zero-shot LLM/VLM agents for 3D scene understanding.** LLMs/VLMs demonstrate exceptional abilities in task reasoning, planning, tool use, and code writing, enabling a new type of AI system (AI agent) that uses LLMs/VLMs to integrate various off-the-shelf modules for 3D scene understanding. In addition to LLM-Grounder [4] and ZS3DVG [3] leveraging LLMs for 3D visual grounding, scene graph-based methods such as OSVG[37] and ConceptGraph[38] focus on building scene graphs to model the relations between objects and search the target object by LLM-parsed query. Recently,

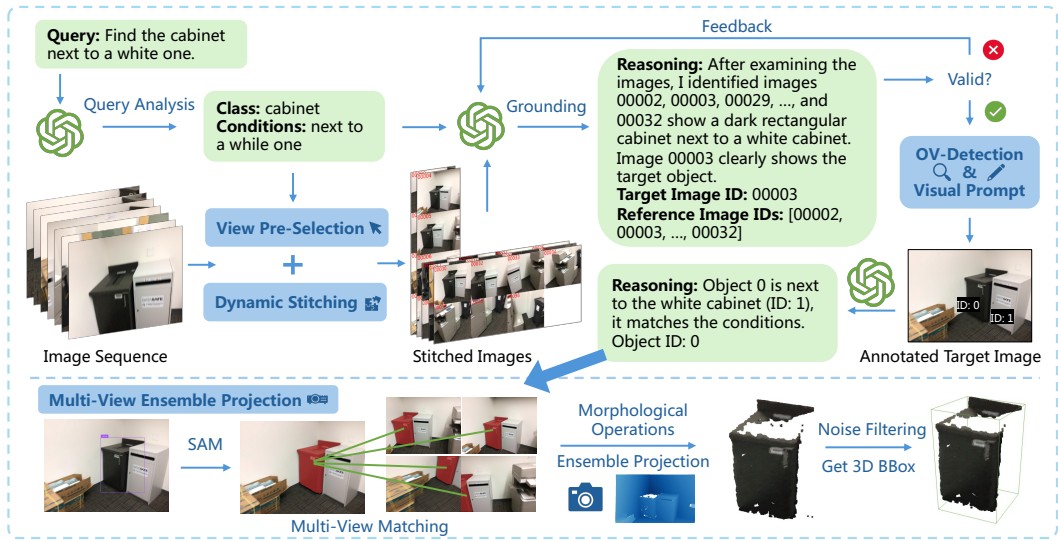

Figure 2: **An overview of VLM-Grounder.** VLM-Grounder analyzes the user query and dynamically stitches image sequences for efficient VLM processing to locate the target image and object. A 2D open-vocabulary detection model and the Segment Anything Model generate a fine-grained mask, which is then projected using a multi-view ensemble strategy to obtain the 3D bounding box.

Agent3D-Zero [39] employs VLMs to understand the bird's-eye view of a 3D scene, retrieving different observational views for tasks like question answering and scene captioning. OpenEQA [40] proposed new question-and-answer datasets to benchmark agents' scene understanding abilities. Different from previous works, our VLM-Grounder focuses on object localization with 3D bounding boxes and directly uses 2D images without requiring a reconstructed 3D scene or 3D localization models. In the video processing domain, VideoAgent [41, 42] and TraveLER [43] also use LLMs for processing 2D images. However, they focus on video event understanding and question answering, rather than scene understanding, and cannot perform 3D localization.

## 3 Methodology

In this section, we present the overall framework of **VLM-Grounder** (Sec. 3.1), and detail the motivations and specifics of three key modules: dynamic stitching (Sec. 3.2), grounding and feedback (Sec. 3.3), and multi-view ensemble projection (Sec. 3.4).

### 3.1 VLM-Grounder

VLM-Grounder processes image sequences of the scanned scene along with a user query to predict the 3D bounding box of the target object. For each scene, we assume access to the intrinsic and extrinsic camera parameters and the depth image for each image. These can be obtained online via RGB-D sensors with (visual-inertial) odometry [44, 45, 46, 47], or RGB-based dense SLAM [48, 49], or offline with SfM [50] and MVS [51]. VLM-Grounder does not depend on reconstructed point clouds or object priors, offering a broader application compared to previous methods.

VLM-Grounder is an agent framework where the VLM is equipped with various tools and modules to enable its grounding capability. In this work, we use GPT-4V as the VLM. Given a user query, each step of VLM-Grounder's process is illustrated in Fig. 2 and described sequentially below.

**Query analysis.** VLM analyzes the query to identify the target class label and grounding conditions.

**View pre-selection and dynamic stitching.** Image sequences scanning the scene are pre-selected using a 2D open-vocabulary detector to retain only those with the target class. These images are then annotated with IDs, stitched, and resized into fewer images using our dynamic stitching strategy.

**Grounding and feedback.** VLM receives the analyzed and original query, and the stitched images to locate the target image and object. If VLM predicts an invalid target, feedbacks are added to the message history, and VLM retries until finding a valid target or reaching the retry limit $M$.

**Open-vocabulary detection and visual prompt.** After the VLM predicts the target image, a 2D open-vocabulary detector detects the image with the target class. If the target image contains multiple instances of the same class, unique IDs are annotated at the center of the detected bounding boxes as visual prompts. VLM then uses these IDs to determine and select the correct target object.

**Multi-view ensemble projection.** The target image and bounding box are input into the Segment Anything Model (SAM) [52] to obtain a fine-grained mask. Other images of the same object are found via image matching, and their masks are also extracted. All masks are post-processed by morphological operations and projected using camera parameters with the depth map to create projected point clouds. These point clouds are filtered for noise to determine the final 3D bounding box.

## 3.2 Dynamic Stitching

Using VLM to process image sequences presents several problems: 1) VLMs have a maximum image limit (*e.g.*, GPT-4V allows only 10 images for Tier-1 users). 2) Inputting many images quickly consumes the VLM's context length, limiting output content and potentially affecting performance. 3) More images increase inference costs, including token usage, latency, and timeout risk.

To address these issues, we conduct view pre-selection to filter images, but this still leaves too many images. Therefore, we stitch multiple images into a single image with a grid layout and resize it according to the VLM's settings. Stitching may lead to information loss, and the chosen layout affects the total number of images sent to the VLM, influencing its understanding of the image sequences. To study the effects of stitching, we designed a novel benchmark called the Visual-Retrieval Benchmark, detailed in Sec. 4.3.

From the benchmark results, we identified the top three layouts for GPT-4V with minimal information loss: (4, 1), (2, 4), and (8, 2), where (4, 1) means 4 rows and 1 column per stitched image. One straightforward approach is to use the best layout (4, 1) as a fixed layout. However, it only accommodates 4 images, which is insufficient for sequences containing many images. To maximize performance, we propose a dynamic stitching strategy that dynamically utilizes the top three layouts.

Specifically, we set a soft limit of the maximum number of stitched images $L$ allowed for the VLM. Given an image sequence with $n$ images, we first attempt to use the (4, 1) layout while keeping the number of stitched images within $L$. If this is not feasible, we stitch some images using larger layouts like (2, 4) and (8, 2). For example, with $n = 40$ and $L = 6$, six stitched images using the (4, 1) layout are insufficient, so we use two (4, 1) and four (2, 4) stitched images. If the total number of images exceeds $(8 \times 2)L$, we exceed the soft limit and use a larger and also effective layout of (9, 3). Pseudo-code for the dynamic stitching strategy is provided in the supplementary material.

## 3.3 Grounding and Feedback

VLM receives the analyzed and original query along with stitched images to identify the image containing the target object. Since determining whether the grounding conditions are met may require considering multiple views, we prompt the VLM to explain its reasoning process and provide the referenced images used.

After the VLM predicts a target image, we check its validity. If the target image does not exist, we append "image-invalid" feedback to the message history and prompt the VLM to reselect. If the target image exists but the 2D open-vocabulary detection model does not detect any objects of the target class, we append "object-not-existing" feedback and prompt the VLM to reselect. When the image and candidate objects are valid, we annotate the target image with different object IDs and prompt the VLM to select the target object ID. If the VLM predicts an invalid object ID, we append "object-ID-invalid" feedback and the VLM should reselect. If the VLM cannot predict a

valid image with a valid object ID after $M$ retries, the process is considered a failure. Details of different feedbacks are provided in the supplementary material.

## 3.4 Multi-View Ensemble Projection

Using a single image for 3D projection may result in incomplete point clouds and low IoU with the ground truth bounding box due to the limited field of view. To address this, we employ multi-view images showing the same target object for joint estimation. We use the image matching method PATS [53] to match the target object mask (anchor) with other images to obtain matched pixel pairs, indicating the same spatial points between images. Using the matching results, these images are processed by a 2D open-vocabulary detector and SAM to get the matched masks. Each mask is projected to obtain its corresponding point clouds. In total, we use $N$ images together.

In scenes with many objects of identical appearance, the image-matching module may produce mismatched results. To filter these mismatched pairs, we calculate the L2 Chamfer Distance of these point clouds with the anchor point clouds and filter out those having a distance larger than 0.1. The final point clouds are the union of these valid point clouds, which are then filtered for noise, and an axis-aligned 3D bounding box is calculated as the final prediction.

It is worth noting that SAM may produce noisy masks, and depth maps may not be accurate, especially at object borders. These issues result in noisy point clouds that cannot be filtered. To address this, each mask undergoes two morphological operations: 1) Erosion to remove noise and shrink the mask border to avoid inaccurate depth at the border. 2) Component selection to retain the top 2 largest connected components of the predicted mask, removing incorrect masks while preserving most of the correct mask. These operations mitigate the effects of over-segmentation and inaccurate depth, improving overall 3D localization accuracy.

# 4 Experimental Results

## 4.1 Experimental Settings

**Datasets.** Following [3], we experiment on the ScanRefer [1] and Nr3D [2] datasets. ScanRefer annotates ScanNet [17] with 51,583 human-written query-target object pairs. Queries are categorized as "Unique", with only one object of the target class in the scene, or "Multiple", with other objects of the same class (distractors) present. The Nr3D dataset, part of ReferIt3D [2], contains 41,503 queries for ScanNet scenes. All target objects in Nr3D have at least one distractor; "Easy" samples have one, while "Hard" samples have two or more. Queries are also classified as "View-Dependent" or "View-Independent" based on the presence of view-dependent relations like "left" or "right". To reduce costs, we randomly select 250 validation samples from each dataset for testing. We report the performance of the baselines from their original papers, and the results on the same 250 validation samples are provided in the supplementary material.

**Evaluation metrics.** The ScanRefer benchmark requires predicting the 3D bounding box of the target object from scene point clouds and queries. Metrics are Acc@0.25 and Acc@0.5, indicating the percentage of samples where the predicted bounding box has an IoU greater than 0.25 or 0.5 with the ground truth. In contrast, the Nr3D benchmark provides ground truth bounding boxes (without class labels) for all objects, focusing on top-1 accuracy in selection. VLM-Grounder does not need such priors for input, so we match our predicted box to the ground truth box with the closest center and use this matched box as our model's prediction.

**Implementation details.** For our experiments, we sample one frame from every 20 frames of the original ScanNet image sequences. We use GPT-4o-2024-05-13 [54] as the VLM, setting the temperature to 0.1 and top_p to 0.3 to balance randomness and creativity. The retry limit is $M = 3$, the image count limit is $L = 6$, and the ensemble image number is $N = 7$. We employ SAM-Huge [52] and Grounding DINO-1.5 [55] as the 2D open-vocabulary detectors. The erosion kernel size is

Table 1: **3D visual grounding results on ScanRefer.** Without using geometric information from point clouds, VLM-Grounder outperforms previous zero-shot methods and achieves performance comparable to supervised learning baselines. * indicates that the evaluation is based on 2D masks.

| Methods | Zero-Shot | w/o. PC | Overall | | Unique | | Multiple | |
|---|---|---|---|---|---|---|---|---|
| | | | Acc@0.25 | Acc@0.5 | Acc@0.25 | Acc@0.5 | Acc@0.25 | Acc@0.5 |
| ScanRefer[1] | ✗ | ✗ | 37.3 | 24.3 | 65.0 | 43.3 | 30.6 | 19.8 |
| TGNN[57] | ✗ | ✗ | 34.3 | 29.7 | 64.5 | 53.0 | 27.0 | 21.9 |
| InstanceRefer[24] | ✗ | ✗ | 40.2 | 32.9 | 77.5 | 66.8 | 31.3 | 24.8 |
| 3DVG-Transformer[18] | ✗ | ✗ | 47.6 | 34.7 | 81.9 | 60.6 | 39.3 | 28.4 |
| BUTD-DETR[28] | ✗ | ✗ | 52.2 | 39.8 | 84.2 | 66.3 | 46.6 | 35.1 |
| OpenScene[10] | ✓ | ✗ | 13.2 | 6.5 | 20.1 | 13.1 | 11.1 | 4.4 |
| LLM-Grounder[4] | ✓ | ✗ | 17.1 | 5.3 | - | - | - | - |
| ZS3DVG[3] | ✓ | ✗ | 36.4 | 32.7 | 63.8 | **58.4** | 27.7 | 24.6 |
| **VLM-Grounder (ours)** | ✓ | ✓ | **51.6** | 32.8 | **66.0** | 29.8 | **48.3** | **33.5** |
| **VLM-Grounder* (ours)** | ✓ | ✓ | **62.4** | **53.2** | **87.2** | **76.6** | **56.7** | **47.8** |

15, and we use Open3D's[56] statistical outlier removal with $nb = 5$ and $std = 1$ for point cloud filtering. All prompts and complete demos for VLM-Grounder are in the supplementary material.

## 4.2 3D Visual Grounding Results

**ScanRefer.** Our VLM-Grounder significantly outperforms all previous zero-shot approaches on the ScanRefer benchmark as shown in Tab. 1. Specifically, it surpasses the previous SOTA method, ZS3DVG[3], by a large margin. For overall Acc@0.25, VLM-Grounder achieves 51.6%, compared to ZS3DVG's 36.4%, reflecting a substantial improvement of 15.2. Even without using point clouds, VLM-Grounder demonstrates superior performance. In contrast, using an open-vocabulary instance segmentation model alone, such as OpenScene[10], results in poor performance (13.2% Acc@0.25), likely because it fails to understand object relationships and relies on a bag-of-words approach for visual grounding [4]. While VLM-Grounder still lags behind one of the SOTA supervised-learning models BUTD-DETR (52.2% Acc@0.25), it achieves comparable performance to earlier baselines like InstanceRefer (40.2%) and 3DVG-Transformer (47.6%) without any training.

Our method shows a notable gap between Acc@0.25 and Acc@0.5. This discrepancy arises because VLM-Grounder operates directly on 2D images and projects the 2D masks into 3D using camera intrinsic and extrinsic parameters along with depth values. These estimated parameters often contain noise, causing inaccuracies in the predicted 3D bounding boxes, *e.g.*, a single outlier can result in an overly large bounding box. Although our multi-view ensemble projection module helps mitigate this issue, it cannot entirely eliminate it. Previous methods rely on reconstructed point clouds and point cloud-based localization models to provide precise object locations, which offer geometric information and bring advantages for evaluation based on 3D bounding box. Nevertheless, VLM-Grounder still outperforms ZS3DVG on the Multiple split for both Acc@0.25 and Acc@0.5.

To further isolate the effects of imperfect projection and better evaluate grounding accuracy, we also assess VLM-Grounder's performance by comparing the IoU of the predicted 2D masks against the ground truth 2D masks. Results show that VLM-Grounder's grounding capability surpasses that of previous zero-shot methods and even outperforms the supervised method BUTD-DETR from a 2D perspective. Additionally, VLM-Grounder exhibits significant improvement in the more challenging Multiple splits, highlighting its superior grounding ability across various scenarios.

**Nr3D.** For the Nr3D benchmark, previous methods use GT 3D bounding boxes as input. This provides an important advantage as it serves as a strong prior. In contrast, VLM-Grounder operates without relying on any object priors or point cloud information, yet still outperforms previous zero-shot methods and even some supervised learning approaches. VLM-Grounder achieves an overall accuracy of 48.0%, surpassing the previous zero-shot SOTA, ZS3DVG, which reaches 39.0%.

Table 2: **3D visual grounding results on Nr3D.** VLM-Grounder surpasses the previous SOTA zero-shot method without requiring access to point clouds or ground-truth bounding box priors.

| Methods | Zero-Shot | w/o. PC | w/o. GT BBox | Overall | Easy | Hard | VD | VID |
|---|---|---|---|---|---|---|---|---|
| ReferIt3D[2] | ✖ | ✖ | ✖ | 35.6 | 43.6 | 27.9 | 32.5 | 37.1 |
| TGNN[57] | ✖ | ✖ | ✖ | 37.3 | 44.2 | 30.6 | 35.8 | 38 |
| InstanceRefer[24] | ✖ | ✖ | ✖ | 38.8 | 46.0 | 31.8 | 34.5 | 41.9 |
| 3DVG-Transformer[18] | ✖ | ✖ | ✖ | 40.8 | 48.5 | 34.8 | 34.8 | 43.7 |
| BUTD-DETR[28] | ✖ | ✖ | ✖ | 54.6 | 60.7 | 48.4 | 46.0 | 58.0 |
| ZS3DVG[3] | ✔ | ✖ | ✖ | 39.0 | 46.5 | 31.7 | 36.8 | 40.0 |
| **VLM-Grounder (ours)** | ✔ | ✔ | ✔ | **48.0** | **55.2** | **39.5** | **45.8** | **49.4** |

The improvement is reflected consistently across various query categories. Without model training, VLM-Grounder's overall performance also competes with supervised learning methods like InstanceRefer (38.8%) and 3DVG-Transformer (40.8%).

### 4.3 Visual-Retrieval Benchmark

Stitching multiple images into one can reduce the number of images input to a VLM, but its impact on the VLM's visual understanding and the existence of optimal layouts are unclear. To explore this, we propose a Visual-Retrieval Benchmark. While our findings are specific to GPT-4V and ScanNet images, the benchmark is general and can be applied to other settings to draw relevant conclusions.

#### 4.3.1 Benchmark Settings

We randomly select 1,000 images from the ScanNet dataset, each annotated with a unique ID. Additionally, a block of random color is generated and placed at a random position within each image. These images are then stitched using various layouts and fed into the VLM, which retrieves all image IDs and the corresponding block colors, as illustrated in Fig. 3. This benchmark allows us to assess the extent of information loss caused by the stitching strategy through retrieval accuracy. We focus primarily on two factors: stitching layouts and the number of images. Additionally, we measure the retrieval time for different numbers of input images. More details are provided in the supplementary material.

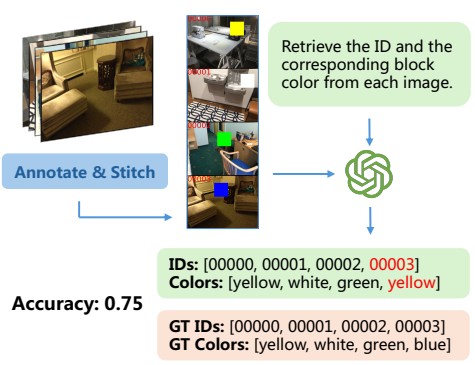

Figure 3: **Visual-Retrieval benchmark.**

#### 4.3.2 Observations

**Stitching layouts.** As shown in Fig. 4(a), we could identify the top three layouts used by VLM-Grounder: (4, 1), (2, 4), and (8, 2), with (4, 1) achieving perfect retrieval. Accuracy significantly declines for layouts denser than (5, 5), suggesting a "resolution" upper bound for effective image stitching. This decline is likely due to GPT-4V's pre-processing step, which resizes images to ensure the long side is less than 2048 pixels and the short side is less than 768 pixels, resulting in lower resolution for each image in denser layouts.

**Number of images.** We use the top five layouts and observe how accuracy varies with the number of images sent to GPT-4V in one request. From Fig. 4(b), increasing the number of images only slightly reduces retrieval accuracy. As shown in Fig. 4(c), the request time increases linearly with the number of images, which is favorable. However, adding more images is not always beneficial because sending too many images (e.g., more than 20) can lead to timeouts and unsuccessful experiments, as indicated by the incomplete results for layouts (3, 3) and (8, 2) and the spikes in Fig. 4(c).

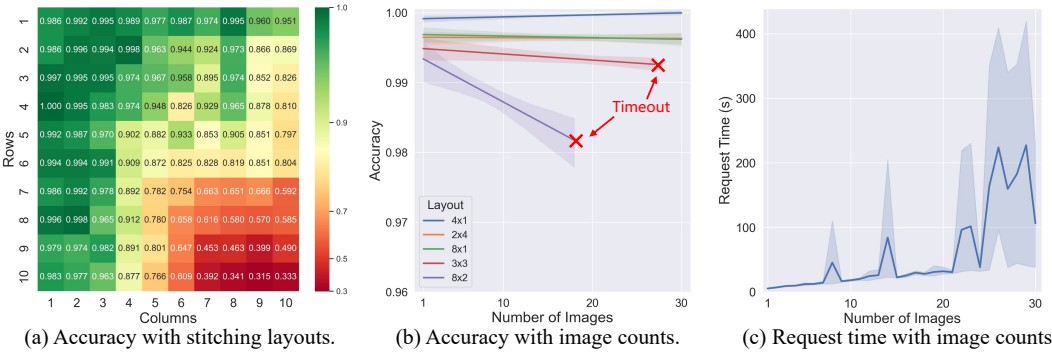

|(a) Accuracy with stitching layouts. | (b) Accuracy with image counts. | (c) Request time with image counts.|

Figure 4: Benchmark accuracy and request time for different stitching layouts and image counts.

Table 3: **Stitching strategies.**

| Strategy | Acc@0.25 |
|---|---|
| Fix (1, 1) | N.A. |
| Fix (8, 2) | 48.4 |
| Square | 49.2 |
| **Dynamic Stitching** | **51.6** |

Table 4: **Number of images.**

| Images | Acc@0.25 |
|---|---|
| **6** | **51.6** |
| 8 | 50.8 |
| 10 | 51.2 |
| 12 | 48.4 |

Table 5: **Projection operations.**

| Operations | Acc@0.25 |
|---|---|
| Baseline | 40.8 |
| +Morpho. Ops | 45.2 |
| +Point Filtering | 48.4 |
| **+Multi-View** | **51.6** |

### 4.4 Ablation Studies

**Stitching strategies.** To validate the effectiveness of our dynamic stitching strategy, we compared it with various stitching approaches: no stitching (1, 1), a fixed layout (8, 2), and a square strategy. The square strategy calculates a stitching layout that approximates a square shape while staying within the image limit. As shown in our results, the proposed dynamic stitching outperforms the others, demonstrating its efficacy. Without stitching, the system often encounters timeouts and fails to complete the task, underscoring the necessity of an effective stitching strategy.

**Image limits.** We experimented with different values for the soft image limit $L$ as shown in Tab. 4. Results indicate that performance remains similar for limits below 10. However, as discussed in Sec. 4.3, increasing the number of images leads to higher inference costs and a greater risk of timeouts. Consequently, we set $L = 6$ in our main experiments to balance performance and efficiency.

**Projection operations.** To assess the impact of different operations within the multi-view ensemble projection module, we incrementally added operations to a baseline and measured their effect. These operations include morphological processing, point cloud filtering, and multi-view ensemble estimation. Tab. 5 shows a clear performance improvement with each additional component, confirming the importance and effectiveness of these operations.

Additional ablations, such as using YOLOv8-World [58] instead of Grounding DINO-1.5[55] as the open-vocabulary detector, are provided in the supplementary material.

## 5 Conclusion and Limitations

In this paper, we presented VLM-Grounder, a VLM agent that excels in zero-shot 3D visual grounding. We introduced a novel Visual-Retrieval benchmark to evaluate the impact of stitching operations on VLM's visual understanding. VLM-Grounder has several appealing properties: it leverages foundation models from the language and 2D domains without training, and offers a more transparent and explainable grounding process than end-to-end models. However, it has limitations. The accuracy of 3D grounding is affected by imprecise camera parameters and depth maps, and targets can be missed if the open-vocabulary detector fails to identify them. Further discussions on limitations, error analysis, inferencing time, and qualitative results are provided in the supplementary material.

**Acknowledgements.** We sincerely thank Tianhe Ren and Lei Zhang from The International Digital Economy Academy (IDEA) for providing access to the Grounding DINO-1.5 model. This research was partially supported by the Centre for Perceptual and Interactive Intelligence (CPII) Ltd. under the Innovation and Technology Commission (ITC)'s InnoHK and Shanghai AI Laboratory.

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

# VLM-Grounder: A VLM Agent for Zero-Shot 3D Visual Grounding

## Supplementary Material

## A  Dynamic Stitching

We employ a dynamic stitching algorithm to organize images into various layouts, with the pseudocode provided in Algorithm 1. The process begins by calculating the largest layout that should be used. Given an image sequence with $n$ images, and a maximum number of stitched images $L$, we first compute the quantity of each layout. We use the variables $n_4$, $n_8$, $n_{16}$, and $n_{27}$ to represent the number of (4, 1), (2, 4), (8, 2), and (9, 3) layouts, respectively.

For example, assuming $n = 84$ and $L = 6$, we know $n \leq 16L$. First, we calculate the minimum number of (8, 2) layouts required. Each (8, 2) layout accommodates 8 more images than a (2, 4) layout, so we divide the number of images exceeding what six (2, 4) layouts can store by 8 to find the minimum number of (8, 2) layouts needed. In this example, it is 5. Next, we compute the layout needed for the remaining images. We update the remaining image count to $84 - 5 * 8 * 2 = 4$ and the stitched image count to $6 - 5 = 1$. Similarly, we determine that we need zero (2, 4) layouts and one (4, 1) layout for the remaining images. Thus, we have determined the number of each layout required. We then generate the stitched images in ascending order of layout size to ensure that only the largest layout may have unused space, thereby minimizing resolution waste.

It is important to note that if the number of images is too large to be accommodated by $L$ images of the largest layout, we select the largest layout to minimize the total number of stitched images. For any excess images, we maximize utilization efficiency by invoking the `dynamic_stitching` function again to find the appropriate layout, setting the fixed number to 1 to minimize the count of stitched images. In this case, we first generate (9, 3) layouts and then recursively call the function to generate the remaining layouts, which may result in some unused space in smaller layouts.

## B  Visual-Retrieval Benchmark Settings

We randomly selected 1,000 images from the ScanNet dataset, assigning each a unique ID ranging from 00000 to 00999. Each image ID was annotated in red at the top-left corner. Additionally, a color block was generated at a random position within each image, using one of six colors: red,

---

**Algorithm 1:** Dynamic Stitching Algorithm

---

**1** **Function** dynamic_stitching($imgs$, $L$)**:**

    // candidate_layouts: (4, 1), (2, 4), (8, 2), (9, 3)

    **Input:** image sequence $imgs$, the maximum number of stitched images $L$

    **Output:** stitched image sequence $res$

**2**     $n \leftarrow \text{len}(imgs)$;

**3**     $res \leftarrow []$;

**4**     **if** $n \leq 4L$ **then**                           // (4, 1) layout is enough

**5**         $res \leftarrow \text{stitch\_image}(imgs, (4, 1))$;

**6**     **else if** $n \leq 8L$ **then**              // at least one (2, 4) layout is used

**7**         $n_8 \leftarrow \lceil (n - 4L)/4 \rceil$;

**8**         $n_4 \leftarrow L - n_8$;

**9**         $res \leftarrow res + \text{stitch\_image}(imgs[0 \ldots 4n_4 - 1], (4, 1))$;

**10**        $res \leftarrow res + \text{stitch\_image}(imgs[4n_4 \ldots ], (2, 4))$;

**11**     **else if** $n \leq 16L$ **then**          // at least one (8, 2) layout is used

**12**        $n_{16} \leftarrow \lceil (n - 8L)/8 \rceil$;

**13**        $n \leftarrow \max(n - 16n_{16}, 0)$ ;             // number of images remaining

**14**        $n_{4,8} \leftarrow L - n_{16}$ ;             // number of (4, 1), (2, 4) layouts

**15**        $n_8 \leftarrow \lceil (n - 4n_{4,8})/4 \rceil$;

**16**        $n_4 \leftarrow n_{4,8} - n_8$;

**17**        $res \leftarrow res + \text{stitch\_image}(imgs[0 \ldots 4n_4 - 1], (4, 1))$;

**18**        $res \leftarrow res + \text{stitch\_image}(imgs[4n_4 \ldots 4n_4 + 8n_8 - 1], (2, 4))$;

**19**        $res \leftarrow res + \text{stitch\_image}(imgs[4n_4 + 8n_8 \ldots ], (8, 2))$;

**20**     **else if** $n \leq 27L$ **then**          // at least one (9, 3) layout is used

**21**        $n_{27} \leftarrow \lceil (n - 16L)/11 \rceil$;

**22**        $n_{4,8,16} \leftarrow L - n_{27}$ ;      // number of (4, 1), (2, 4), (8, 2) layouts

**23**        $n \leftarrow \max(n - 27n_{27}, 0)$ ;             // number of images remaining

**24**        $n_{16} \leftarrow \lceil (n - 8n_{4,8,16})/8 \rceil$;

**25**        $n_{4,8} \leftarrow n_{4,8,16} - n_{16}$ ;           // number of (4, 1), (2, 4) layouts

**26**        $n \leftarrow \max(n - 16n_{16}, 0)$;

**27**        $n_8 \leftarrow \lceil (n - 4n_{4,8})/4 \rceil$;

**28**        $n_4 \leftarrow n_{4,8} - n_8$;

**29**        $res \leftarrow res + \text{stitch\_image}(imgs[ 0 \ldots 4n_4 - 1], (4, 1))$;

**30**        $res \leftarrow res + \text{stitch\_image}(imgs[4n_4 \ldots 4n_4 + 8n_8 - 1], (2, 4))$;

**31**        $res \leftarrow res + \text{stitch\_image}(imgs[4n_4 + 8n_8 \ldots 4n_4 + 8n_8 + 16n_{16} - 1], (8, 2))$;

**32**        $res \leftarrow res + \text{stitch\_image}(imgs[4n_4 + 8n_8 + 16n_{16} \ldots ], (9, 3))$;

**33**     **else**                                 // use more than $L$ stitched images

**34**        $n_{27} \leftarrow \lfloor n/27 \rfloor$;

**35**        $res \leftarrow res + \text{stitch\_image}(imgs[ 0 \ldots 27n_{27} - 1], (9, 3))$;

**36**        $res \leftarrow res + \text{dynamic\_stitching} (imgs[27n_{27} \ldots ], 1)$;

**37**     **return** $res$;

---

green, blue, yellow, white, or black. The images were then stitched using specific layouts, forming the basic image sets sent to the VLM. The VLM's task was to identify all images, retrieve their IDs, and determine the color of the blocks. The VLM was required to return two lists—IDs and corresponding colors—as demonstrated in Fig.3. of the main paper.

Occasionally, the VLM might retrieve the same ID from different images, leading to conflicts where multiple ID-color pairs exist for the same ID. In such cases, if at least one retrieved ID matches the ground truth, it is considered correct. In other words, we calculated the Recall as the accuracy in this benchmark. For instance, in Fig.3. of the main paper, if four images were input and the VLM retrieved four pairs, but the pair 00003-yellow was incorrect (the ground truth being 00003-blue), the accuracy for this benchmark would be 0.75.

The benchmark investigated two primary variables:

**Stitching layout.** The stitching layout defines the rows and columns in which images are stitched, which can be regarded as "visual resolution".

**Visual length.** The number of stitched images included in a single conversation, which can be regarded as "visual context length".

We also measured the request time cost. By duplicating an image from 1 to 30 times within a request, we conducted 10 trials for each duplication count and calculated the average request time cost.

## C VLM-Grounder Prompts

We used several prompts in our work, as shown in the Tab. 6, including query_analysis_prompt, grounding_system_prompt, input_prompt, bbox_select_prompt, image_ID_invalid_prompt, and detection_not_exist_prompt.

For each query, we utilize the query_analysis_prompt to extract the category and associated conditions of the target object, such as position, shape, color, or relative relationships with other objects. In the grounding and feedback process, we first employed the grounding_system_prompt to guide VLM in performing visual grounding tasks. Then, we utilize the input_prompt to provide information such as our image, query statement, target object category, and grounding conditions, with stitched images appended. VLM would return the query results in the specified JSON format.

If the target image ID in the returned results does not contain any target object, we use the detection_not_exist_prompt to inform VLM and request it to make a new selection. In case the image ID provided cannot find the corresponding image, we employ the image_ID_invalid_prompt to notify VLM for a fresh selection. Furthermore, if there are multiple target objects in the chosen image, we use the bbox_select_prompt to instruct VLM in selecting the correct bounding box ID.

Table 6: **Prompts of VLM-Grounder.** The placeholders in the table represent different variables. {query} denotes the user query, while {pred_target_class} and {conditions} represent the target object's category and grounding conditions, respectively. {num_view_selections} refers to the total number of images, and {num_candidate_bboxes} indicates the number of candidate bounding boxes. In the image_ID_invalid_prompt and detection_not_exist_prompt, {image_id} refers to the image ID selected by the VLM.

| |
| --- |
| **query_analysis_prompt** |

You are working on a 3D visual grounding task, which involves receiving a query that specifies a particular object by describing its attributes and grounding conditions to uniquely identify the object. Here, attributes refer to the inherent properties of the object, such as category, color, appearance, function, etc. Grounding conditions refer to considerations of other objects or other conditions in the scene, such as location, relative position to other objects, etc. Now, I need you to first parse this query, return the category of the object to be found, and list each of the object's attributes and grounding conditions. Each attribute and condition should be returned individually. Sometimes the object's category is not explicitly specified, and you need to deduce it through reasoning. If you cannot deduce after reasoning, you can use 'unknown' for the category. Your response should be formatted as a JSON object. Here are some examples:

Input:

Query: this is a brown cabinet. it is to the right of a picture.

Output:

{

"target_class": "cabinet",

"attributes": ["it's brown"],

"conditions": ["it's to the right of a picture"]

}

...(two more examples)

Ensure your response adheres strictly to this JSON format, as it will be directly parsed and used.

Query: {query}

---

**grounding_system_prompt**

---

You are good at finding objects specified by user queries in indoor rooms by watching the videos scanning the rooms.

---

**bbox_select_prompt**

---

Great! Here is the detailed version of your selected image. There are {num_candidate_bboxes} candidate objects shown in the image. I have annotated each object at the center with an object ID in white color text and black background. Do not mix the annotated IDs with the actual appearance of the objects. Please give me the ID of the correct target object for the query. Reply using JSON format with two keys "reasoning" and "object_id" like this:

{

"reasoning": "your reasons", // Explain the justification why you select the object ID.

"object_id": 0 // The object ID you selected. Always give one object ID from the image, which you are the most confident of, even you think the image does not contain the correct object.

}

---

**image_ID_invalid_prompt**

---

The image {image_id} you selected does not exist. Did you perhaps see it incorrectly? Please reconsider and select another image. Remember to reply using JSON format with the three keys "reasoning", "target_image_id", and "reference_image_ids" as required before.

---

**detection_not_exist_prompt**

---

The image {image_id} you selected does not seem to include any objects that fall into the category of {pred_target_class}. Please reconsider and select another image. Remember to reply using JSON format with the three keys "reasoning", "target_image_id", and "reference_image_ids" as required before.

---

---

**input_prompt**

---

Imagine you are in a room and are asked to find one object. Given a series of images from a video scanning an indoor room and a query describing a specific object in the room, you need to analyze the images to locate the object mentioned in the query within the images. You will be provided with multiple images, and the top-left corner of each image will have an ID indicating the order in which it appears in the video. Adjacent images have adjacent IDs. Please note that to save space, multiple images have been combined into one image with dynamic layouts. You will also be provided with a query sentence describing the object that needs to be found, as well as a parsed version of this query describing the target class of the object to be found and the conditions that this object must satisfy. Please find the ID of the image containing this object based on these conditions. Note that I have filtered the video to remove some images that do not contain objects of the target class. To locate the target object, you need to consider multiple images from different perspectives and determine which image contains the object that meets the conditions. Note, that each condition might not be judged based on just one image alone. Also, the conditions may not be accurate, so it's reasonable for the correct object not to meet all the conditions. You need to find the most possible object based on the query. If you think multiple objects are correct, simply return the one you are most confident of. If you think no objects are meeting the conditions, make a guess to avoid returning nothing. Usually the correct object is visible in multiple images, and you should return the image in which the object is most clearly observed. Your response should be formatted as a JSON object with three keys "reasoning", "target_image_id", and "reference_image_ids" like this:
{
"reasoning": "your reasoning process" // Explain the process of how you identified and located the target object. If reasoning across different images is needed, explain which images were used and how you reasoned with them.
"target_image_id": "00001", // Replace with the actual image ID (only one ID) annotated on the image that contains the target object.
"reference_image_ids": ["00001", "00002", ...] // A list of IDs of images that are used to determine wether the conditions are met or not.
}
Here is a good example:
query: Find the black table that is surrounded by four chairs.
{
"reasoning": "After carefully examining all the input images, I found image 00003, 00005, and 00021 contain different tables, but only the tables in image 00003 and 00021 are black. Further, I found image 00001, image 00002, image 00003, and image 00004 show four chairs and these chairs surround the black table in image 00003. The chair in image 00005 does not meet this condition. So the correct object is the table in image 00003",
"target_image_id": "00003",
"reference_image_ids": ["00001", "00002", "00003", "00004"]
}
Now start the task:
Query: "{query}"
Target Class: {pred_target_class}
Conditions: {conditions}
Here are the {num_view_selections} images for your reference.

---

# D   More Results and Analyses

## D.1   Ablation on 2D Detectors

As Grounding DINO-1.5 [55] is a closed-source model, we can only request detections through its API. For open-source research, we also employ the widely-used open-source alternative YOLOv8-World [58, 59] for our experiments. Results on the ScanRefer [1] dataset are presented in Tab. 7.

## D.2   Results on Selected 250 Samples

We reproduced previous zero-shot methods and the supervised-learning method BUTD-DETR[28], evaluating their performances using the same 250 validation samples from ScanRefer[1] as VLM-Grounder, with their official codebases. The results are shown in Tab. 8. We use ground-truth bounding boxes for ZS3DVG[3], which produces the upper bound results. Using the same evaluation

Table 7: **3D visual grounding results with YOLOv8-World and Grounding DINO 1.5.** * indicates that the evaluation is based on 2D masks.

| Methods | Overall | | Unique | | Multiple | |
|---|---|---|---|---|---|---|
| | Acc@0.25 | Acc@0.5 | Acc@0.25 | Acc@0.5 | Acc@0.25 | Acc@0.5 |
| VLM-Grounder (YOLOv8-World) | 44.8 | 28.4 | 57.5 | 31.9 | 41.9 | 27.6 |
| VLM-Grounder (GDINO-1.5) | 51.6 | 32.8 | 66.0 | 29.8 | 48.3 | 33.5 |
| VLM-Grounder* (YOLOv8-World) | 53.2 | 45.2 | 74.5 | 63.8 | 48.3 | 40.9 |
| VLM-Grounder* (GDINO-1.5) | 62.4 | 53.2 | 87.2 | 76.6 | 56.7 | 47.8 |

Table 8: **Baseline results on the selected 250 samples.**

| Methods | Overall | | Unique | | Multiple | |
|---|---|---|---|---|---|---|
| | Acc@0.25 | Acc@0.5 | Acc@0.25 | Acc@0.5 | Acc@0.25 | Acc@0.5 |
| BUTD-DETR[28] | 54.0 | 38.4 | 80.9 | 61.7 | 47.8 | 33.0 |
| OpenScene[10] | 18.8 | 5.2 | 27.2 | 7.5 | 0.0 | 0.0 |
| LLM-Grouner[4] | 12.0 | 4.4 | 12.1 | 4.0 | 11.7 | 5.2 |
| ZS3DVG[3] | 31.2 | 31.2 | 55.3 | 55.3 | 25.6 | 25.6 |
| **VLM-Grounder (ours)** | **51.6** | **32.8** | **66.0** | 29.8 | **48.3** | **33.5** |

data, we can verify that our VLM-Grounder outperforms previous zero-shot methods and achieves comparable performance to one of the SOTA supervised-learning methods.

## D.3 Inference Time

We calculated the average processing time of each modules, detailed in Tab. 9. The average processing time per sample is 38.3 seconds or 50.3 seconds, depending on whether VLM needs to be queried again to select the target instance, i.e., when the selected image contains multiple instances of the same categories.

It's worth noting that our work focuses on building a research prototype and verifying its effectiveness in solving the zero-shot 3D visual grounding problem. The processing time has the potential to be significantly improved for the following reasons:

**Implementation improvements.** As we focus on building a research prototype, we prioritize easy-to-understand implementation over cost-optimized implementation. For example, the dynamic-stitching costs about 33% of the total processing time because we use the matplotlib library, which requires initializing a plotting canvas and plotting and annotating the images one by one. Avoiding the use of matplotlib could reduce this time.

**Local deployment of VLMs.** Currently, we use OpenAI's APIs for query analysis, image selection, and instance selection, which involves sending texts and images over the internet, resulting in delays due to network speed. However, deploying VLMs locally on edge devices or robots is an active research direction with promising results from industry efforts. We expect that in future deployment of VLM-Grounder, local VLMs can be used to significantly reduce processing time.

**Efficient 2D foundation models.** Although we use 2D foundation models, they do not necessarily introduce processing time bottlenecks, as there are capable models optimized for efficiency. For example, the open-vocabulary 2D detection model Yolov8-world can run in real time. In this work, we use SAM-Huge for image segmentation for better performance, which takes 0.6 seconds to process one image. However, we have tried using SAM-Base, which takes only 0.2 seconds per image with minimal performance sacrifice (31.6 vs. 32.8 overall acc@0.5 on ScanRefer).

Table 9: **Inference time of different modules (unit: seconds).**

| Query Analysis | View-Preselection | Dynamic Stitching | Img. Selection by VLM |
|---|---|---|---|
| 1.1 | 1.1 | 12.8 | 16 |
| OV Detection | Ins. Selection by VLM (optional) | Img. Seg. | Image Matching |
| 0.1 | 12 | 0.6 | 0.4 |
| Ens. Ims. Seg. (7 imgs) | Projection | Outlier Removal | Overall |
| 5 | 0.4 | 0.8 | 38.3 or 50.3 |

Table 10: **Success rates of different modules.**

| Query Analysis | View Pre-Selection | Image Selection by VLM | OV-Detection |
|---|---|---|---|
| 100% | 96% | 77% | 92% |
| Instance Selection by VLM | Image Segmentation | Multi-View Ensemble Projection | Overall Acc@0.5 |
| 100% | 82% | 61% | 34% |

## D.4 Success Rates and Error Analysis

We randomly sampled 50 samples from the ScanRefer evaluation data and manually inspected the results of each step across the framework. The success rates are shown in Tab. 10. The standards for each step to be regarded as successful are illustrated as follows:

- **Query analysis.** The analyzed query correctly identifies the target category and conditions.
- **View pre-selection.** The pre-selected views contain the target object.
- **Image selection by VLM.** The selected view contains the target object.
- **OV-detection.** The detection results contain the correct detections of the target category.
- **Instance selection by VLM.** The target object is selected.
- **Image segmentation:** The instance mask of the target object predicted by SAM is neither over-segmented nor under-segmented.
- **Multi-view ensemble projection:** The IoU3D of the predicted bounding box and the GT bounding box is greater than or equal to 0.5.

We found that the main errors occur in the grounding (image selection by VLM), OV-detection, image segmentation, and projection modules. We provide more detailed error analyses and visualizations of these errors.

**VLM grounding module.** Typical failure cases of the VLM grounding module are listed below, and the corresponding illustrations are in Fig. 5.

- **Incorrect condition analysis.** In Fig. 5(1), the VLM is tasked with finding the table between the rug and the carpet. However, it fails to consider all conditions and finds the table on the carpet.
- **Misidentification of the target.** In Fig. 5(2), the VLM is asked to identify a black keyboard in front of a monitor. It mistakenly identifies the box in image 00017 as a keyboard and overlooks the positional context.
- **Ambiguous query descriptions.** There are queries in the ScanRefer dataset that specify objects using view-dependent relations such as left or right, but these queries do not specify the view direction. In such cases, it's difficult for the VLM to correctly find the target object. In Fig. 5(3), the query states that the chair is on the right side of the table, which is insufficient to locate the target due to its ambiguity from different viewpoints.

**Open-vocabulary detection module.** The open-vocabulary detection module may fail due to incorrect semantic analysis, as shown in Fig. 6. It's possible that the training data in such detection models lack corresponding samples. For example, there are more samples with toilets but very few with toilet flush buttons.

**Image segmentation module.** Fig. 7 illustrates the failure cases of the SAM (Segment Anything Model) module. SAM occasionally over-segments instances due to similar textures. For example, in Fig. 7(1), SAM over-segments the shadow of the pillow, and in Fig. 7(3), it over-segments the table near the target chair. Additionally, in Fig. 7(3), SAM under-segments the target chair, missing the chair legs under the table—again due to similar textures. Sudden changes in color can also lead to under-segmentation, as demonstrated in Fig. 7(2), where the part of the curtain with bright lighting is missed.

**Projection module.** In Fig. 8, we present a typical failure case of the projection module, which is caused by an inaccurate depth map. As shown in Fig. 8(1), the depth of the chair edge is incorrectly estimated, leading to the projection of the pixels at the edge turning into a long "tail" (as shown in Fig. 8(2)) in the point cloud. This results in the predicted 3D bounding box (red bounding box in the figure) being larger than the ground-truth bounding box (green bounding box in the figure). Such outliers are difficult to remove using outlier removal algorithms because they are numerous. We also attempted to use clustering methods like DBSCAN to filter these outliers, but since they are connected with inliers, the clustering algorithm may either produce clusters that are too large (as shown in Fig. 8(3)) or too small (as shown in Fig. 8(4)). We adopted image morphological operations to shrink the instance mask, which significantly alleviated the problem but could not completely eliminate it. Such noisy point clouds may also come from inaccurate camera poses.

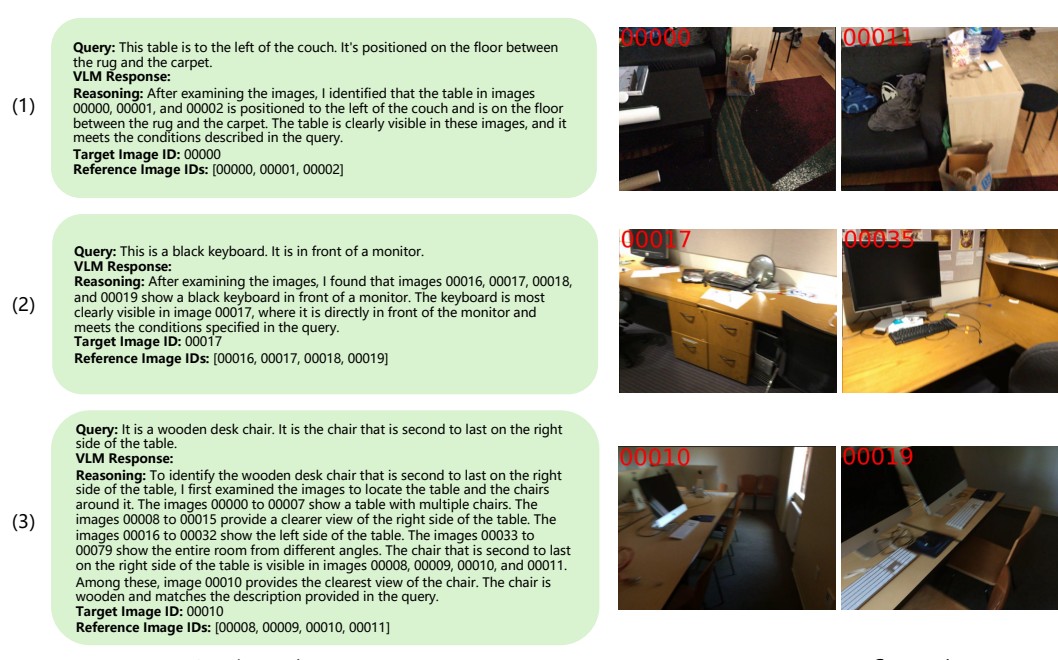

Figure 5: **Failure cases of the VLM grounding module.**

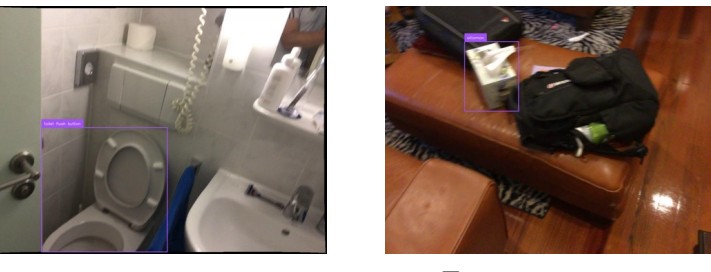

**Target:** toilet flush button

(1)

**Target:** ottomon

(2)

Figure 6: **Failure cases of the open vocabulary detection module.**

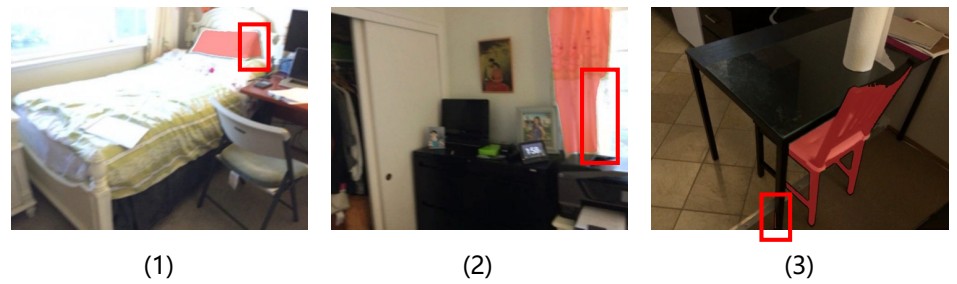

(1)                    (2)                    (3)

Figure 7: **Failure cases of the SAM module.**

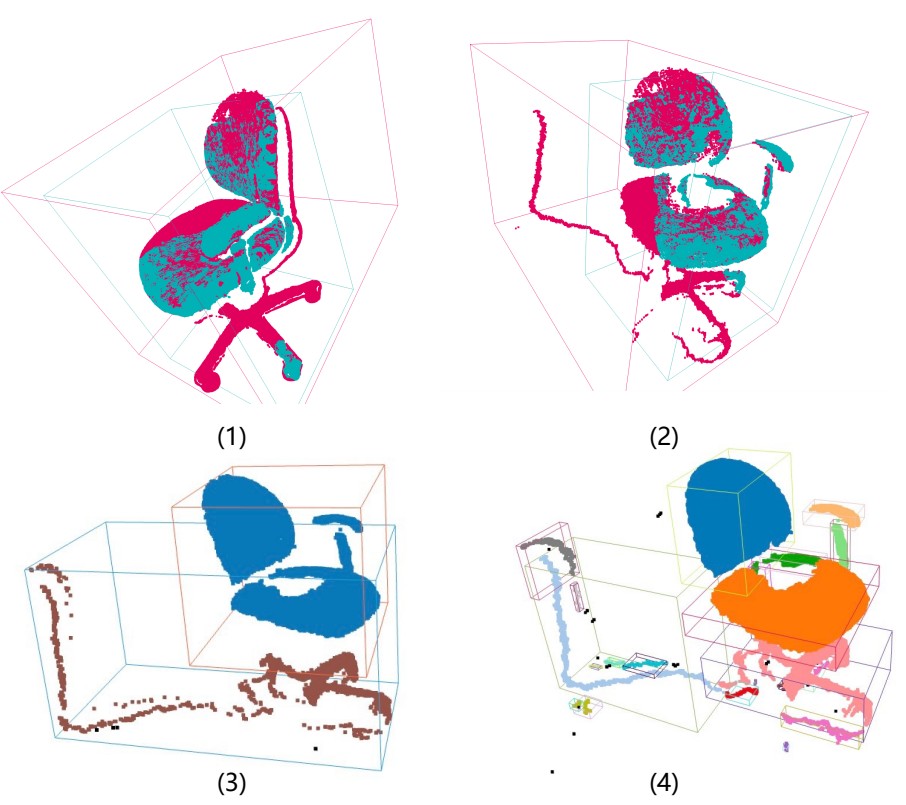

(1)                                        (2)

(3)                                        (4)

Figure 8: **A failure case of the projection module.**

### D.5 Summary of Limitations

While VLM-Grounder achieves superior zero-shot 3D visual grounding by directly operating on 2D images without requiring 3D point clouds or object priors, it has several limitations:

**Capabilities of VLMs.** VLM-Grounder depends on the vision-language model (VLM) for analyzing grounding conditions and locating target objects in sequences of 2D images. If the VLM lacks the ability to process multiple images or struggles with scene understanding from real 2D scans, performance may degrade. In this study, we use the GPT-4o model, which delivers excellent results. VLM technology is continuously advancing, and VLM-Grounder's modular design allows us to replace the current VLM with more powerful models as they become available, potentially enhancing future performance.

**Noise from 2D models.** VLM-Grounder utilizes off-the-shelf 2D open-vocabulary detectors and segmentation models to filter images and generate detailed image masks for projection. Despite their strengths, these 2D foundation models are not infallible. Issues like missed detections, false detections, or incorrect segmentations can prevent VLM-Grounder from identifying the target object, lead to selecting the wrong object, or produce noisy target masks. This noise can result in inaccurate 3D bounding box projections.

**Noise from sensors.** VLM-Grounder predicts the 3D bounding box of the target object from 2D images, relying on accurate camera intrinsics, extrinsics, and depth maps. However, in datasets like ScanNet [17], these parameters often contain noise. For instance, depth sensors can be inaccurate at object boundaries, and RGB images may suffer from motion blur. Such sensor noise leads to inaccuracies in the predicted 3D bounding boxes. While sensor noise is an unavoidable challenge in robotic vision, VLM-Grounder attempts to mitigate these issues through its grounding and feedback scheme combined with multi-view ensemble projection. However, it cannot completely eliminate the effects of sensor inaccuracies. In practical robotic deployments, robots typically have multiple types of sensors. Using multi-sensor fusion can help reduce noise and improve VLM-Grounder's performance.

### D.6 Full Demos

In this section, we present three demonstrations to elucidate the capabilities and behavior of VLM-Grounder in various scenarios. First, in Fig. 9, we illustrate the basic execution process involving a single target object within a scene. Subsequently, we demonstrate the execution process in a more complex scene containing multiple target objects, where the VLM is employed to accurately select the correct object, as in Fig. 10. Lastly, we showcase the execution process in a scenario where the VLM initially selects an incorrect image, thereby triggering a feedback mechanism, as shown in Fig. 11. Morphological operations are applied to all the masks including matched images. In all these examples, we only illustrate four ensemble images and show the result of the morphological operation on the anchor mask. The system_prompt and query analysis are also omitted in the figures for clarity.

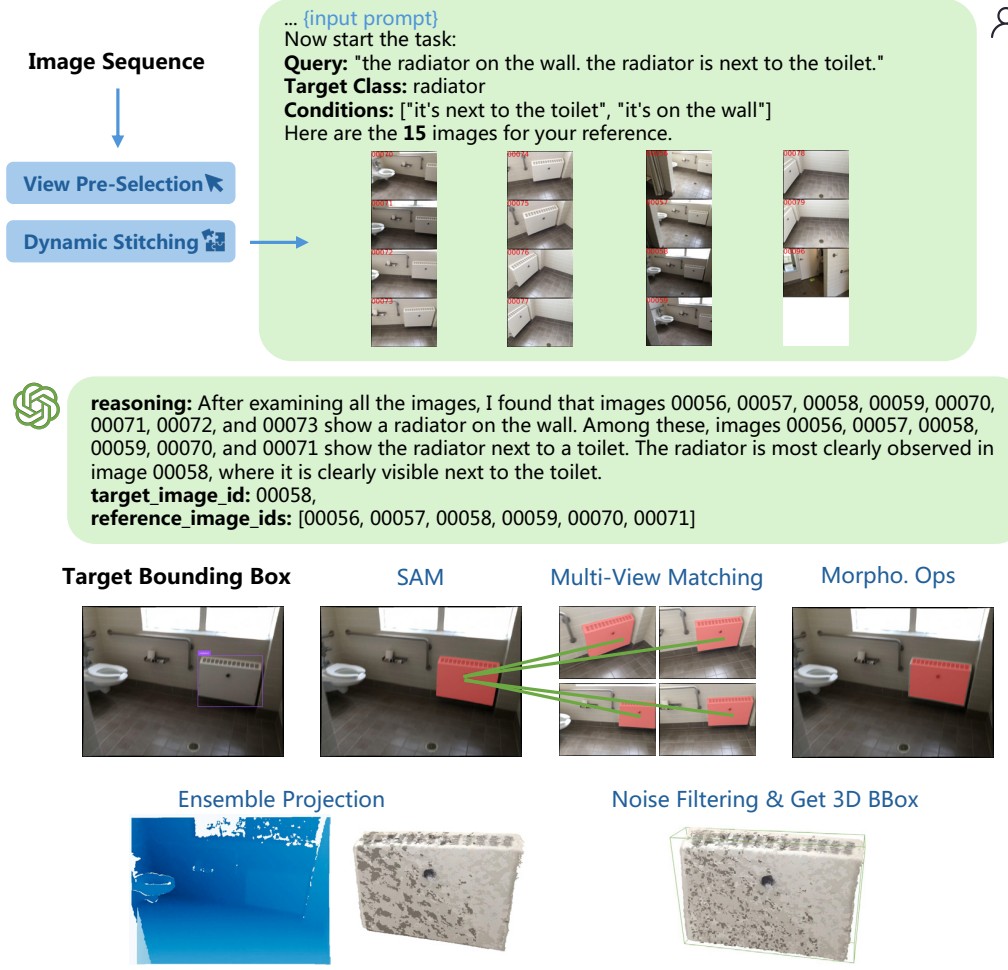

Figure 9: **Demo of VLM-Grounder.**

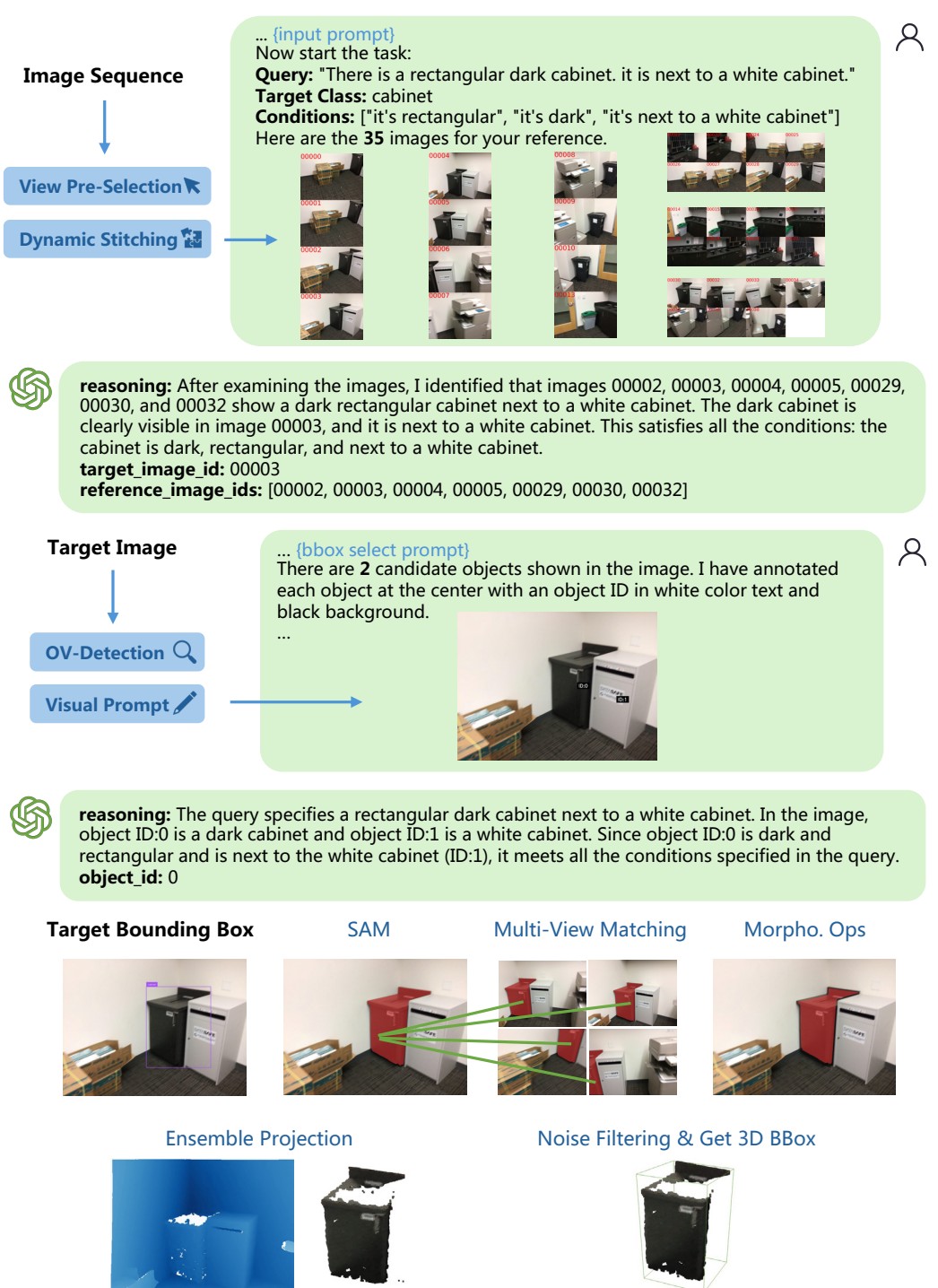

Figure 10: **Demo of VLM-Grounder with several target objects in the scene.**

**Image Sequence**

↓

**View Pre-Selection** ↖

**Dynamic Stitching** 🧩 →

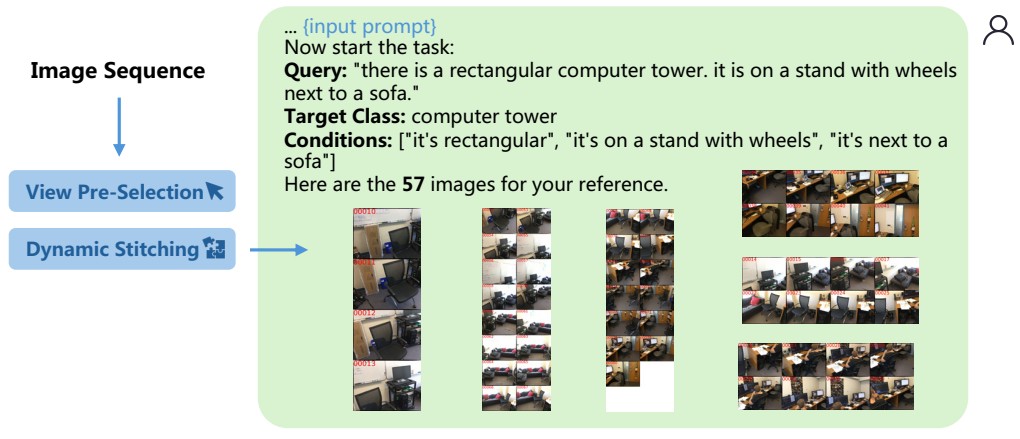

… {input prompt}
Now start the task:
**Query:** "there is a rectangular computer tower. it is on a stand with wheels next to a sofa."
**Target Class:** computer tower
**Conditions:** ["it's rectangular", "it's on a stand with wheels", "it's next to a sofa"]
Here are the **57** images for your reference.

**reasoning:** After examining the images, I found that images 00013, 00014, 00015, 00016, 00017, 00055, 00056, 00057, 00058, 00059, 00060, 00061, 00062, and 00063 show a computer tower that is rectangular and on a stand with wheels. Additionally, images 00016, 00017, 00060, 00061, 00062, and 00063 show that this stand is next to a sofa. Among these, image 00061 provides the clearest view of the computer tower on the stand with wheels next to the sofa.
**target_image_id:** 00061
**reference_image_ids:** [00013, 00014, 00015, 00016, 00017, 00055, 00056, 00057, 00058, 00059, 00060, 00061, 00062, 00063]

The image **61** you selected does not seem to include any objects that fall into the category of computer tower. Please reconsider and select another image. Remember to reply using JSON format with the three keys "reasoning", "target_image_id", and "reference_image_ids" as required before.

**reasoning:** Upon re-evaluating the images, I found that images 00013, 00014, 00015, 00016, 00017, 00055, 00056, 00057, 00058, and 00059 show a rectangular computer tower on a stand with wheels. Additionally, images 00016 and 00017 show that this stand is next to a sofa. Among these, image 00016 provides the clearest view of the computer tower on the stand with wheels next to the sofa.
**target_image_id:** 00016
**reference_image_ids:** [00013, 00014, 00015, 00016, 00017, 00055, 00056, 00057, 00058, 00059]

**Target Bounding Box**  **SAM**  **Multi-View Matching**  **Morpho. Ops**

**Ensemble Projection**  **Noise Filtering & Get 3D BBox**

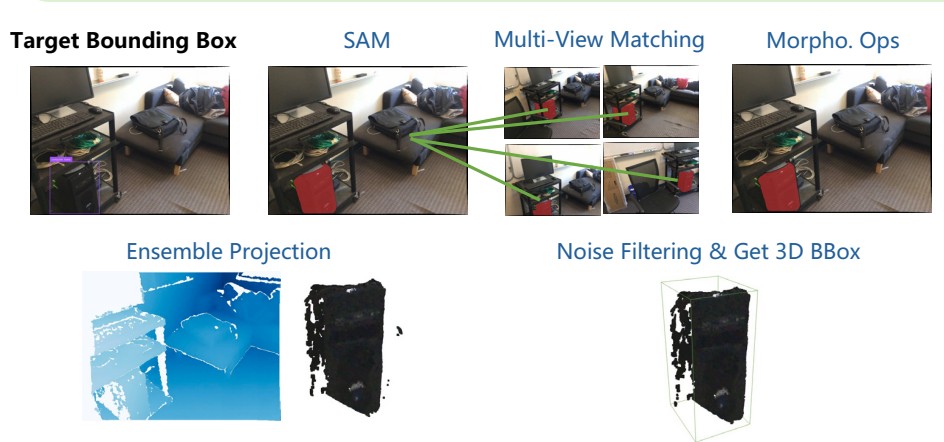

Figure 11: **Demo of VLM-Grounder with feedback.**

