# OpenReview forum: "VLM-Grounder: A VLM Agent for Zero-Shot 3D Visual Grounding"
_robot-learning.org/CoRL/2024/Conference — CoRL 2024_

### Official Review · Reviewer_oE1Z · 2024-07-20
**Need improvement on literature review and experiment**

**Originality:** 2
**Technical Quality:** 3
**Clarity Of Presentation:** 4
**Potential Impact:** 2
**Recommendation:** 2
**Confidence:** 3

**Review:**

Yet another LLM/VLM-based agent work, as we have seen many during the past year. Given the inherent power of LLMs and VLMs, agent work must clearly articulate what they add to the capabilities of LLMs or VLMs beyond simply invoking them or writing basic APIs.

From this perspective, the primary contributions of this paper are "view pre-selection+dynamic stitching," and "multi-view matching." The first step is used to save input tokens, while the second step filters out outliers and generates a better point cloud. The authors conduct experiments on these ablations and show that these components result in an improvement of about 2-3% each. However, "multi-view matching" is also very common in 2D-3D works, which brings the novelty of this paper into question.

Furthermore, VLM/LLM grounding is a highly crowded field today. As far as I know, another stream of research combines LLMs with open-vocabulary scene graphs, such as OVSG and ConceptGraph. These methods are designed for 3D grounding problems and use scene graphs to understand object-level relationships. The authors need to conduct a more thorough literature review and experimental comparison to position their work within the current research landscape.

**Quality Of The Limitations Section:**

3

**Questions For Rebuttal:**

1. The author needs to further discuss their unique contributions and insights on top of the VLM. Given the strength of models like GPT-4 and SAM, the contribution and novelty of this work appear limited.

2. The literature review on 3D grounding from language is insufficient. Works that are scene-graph based, such as OVSG, ConceptGraph, and CLIP-based methods like CLIP-Field, should also be acknowledged and potentially compared.

**Robotics Focus:**

3

**Summary Of Paper:**

This paper presents a new grounding method using VLM.

**Summary Of Recommendation:**

The novelty of this work is limited and require more experiments and literature review.

---

### Official Review · Reviewer_mxB2 · 2024-07-21
**VLM-Grounder: A VLM Agent for Zero-Shot 3D Visual Grounding**

**Originality:** 4
**Technical Quality:** 3
**Clarity Of Presentation:** 4
**Potential Impact:** 4
**Recommendation:** 3
**Confidence:** 5

**Review:**

In this paper, the authors present a novel framework for 3D visual grounding. They utilize a pretrained VLM (ChatGPT-4v), which processes multiple stitched images and a text prompt to locate the target object within the images. This object is then annotated using Grounding-DINO1.5, followed by a selection process via the VLM to identify the correct object. Subsequently, the object is segmented using SAM, and a multi-view ensemble process calculates the 3D bounding box at the end.

Below are my comments about this paper:

My primary concern with this paper is the lack of comprehensive error analysis across the multiple layers and steps within the framework. While there is an error analysis provided for the multi-view ensemble projection step, showcased in Table 1 as VLM-Grounder vs VLM-Grounder*, which illustrates the error percentage from imperfections in the bounding box projection process, similar analyses are absent for other crucial steps such as query analysis, view pre-selection, grounding, human feedback, ov-detection, and visual prompt. It is essential to include error analyses for these stages as well, to experimentally support the rationale behind each layer.

The human feedback layer leaves me somewhat confused. Although it seems like a logical inclusion, I question why this step cannot be entirely automated by validating the image using a 2D open vocab detector, especially since such a layer exists after human validation. This concern ties directly to my previous point, as a comparative analysis of the accuracy with and without human validation would show the significance of this step.

The rationale behind dynamic stitching and visual retrieval benchmarks is well-articulated, and the interpretations of the experimental results are informative, enhancing the overall understanding of the paper.

However, an odd outcome is shown where smaller patches exhibit less accuracy than larger ones, particularly for (1,1) patches, which indicates that each image is queried individually. If the authors based these statistics on a single trial, it would be advisable to conduct multiple runs and compute an average to ensure the reliability of the results. If that is not the case, there is currently no explanation for this outcome.

Minor Comments:

Row 136: Change "like" to "such as" for a more formal expression.

Section 4.3.2 title: Change "Observations." to "Observations" for consistency in formatting.

**Quality Of The Limitations Section:**

3

**Questions For Rebuttal:**

please address the following issues:

- error analysis for each layer
- human feedback step
- patch size accuracy inconsistency

which are expressed in detail in the review above.

**Robotics Focus:**

3

**Summary Of Paper:**

In this paper, the authors present a novel framework for 3D visual grounding. They utilize a pretrained VLM (ChatGPT-4v), which processes multiple stitched images and a text prompt to locate the target object within the images. This object is then annotated using Grounding-DINO1.5, followed by a selection process via the VLM to identify the correct object. Subsequently, the object is segmented using SAM, and a multi-view ensemble process calculates the 3D bounding box at the end.

**Summary Of Recommendation:**

I recommend this paper overall, assuming the concerns will be answered during the rebuttal, it has a good impact on vision and robotics combined research.

---

### Official Review · Reviewer_oazi · 2024-07-25

**Originality:** 2
**Technical Quality:** 2
**Clarity Of Presentation:** 3
**Potential Impact:** 2
**Recommendation:** 2
**Confidence:** 4

**Review:**

# Strength
1. VLMs possess powerful generalization capabilities and utilizing 2D VLM to tackle 3D grounding is a promising solution.
2. The proposed pipeline is reasonable.
3. The paper is easy to follow.

# Weakness
1. The biggest concern is the results in Table 1. As mentioned in line 183 in Sec 4.1, the experiment is conducted on 250 validation samples for VLM-Grounder. However, the results in Table 1 of other methods are obtained by evaluating either the whole validation set of ScanRefer or a different subset of ScanRefer, which are both different from the 250 samples. So, simply comparing the performance is not fair, which makes the conclusion untrustworthy.
2. Due to the participation of foundation models, an average processing time for each sample should be provided.
3. It would be better if the authors could provide the success rate and a detailed analysis of the failure cases, which can provide more insights into the capabilities of the model.
4. Lack of the analysis of using PATS for matching the target object mask with other images, e.g., recall and precision.

**Quality Of The Limitations Section:**

3

**Questions For Rebuttal:**

1. The evaluation data must be unified across the methods in Table 1, and experiments in Table 1 must be re-implemented to make a fair comparison with other methods.
2. What’s the average inference speed for each sample?
3. The notable gap between Acc@0.25 and Acc@0.5 proves that explicit 3D representation such as point cloud is probably necessary for generating accurate 3D results. Maybe the authors need to clarify this point.

**Robotics Focus:**

3

**Summary Of Paper:**

This paper proposes VLM-Grounder, a novel method for 3D visual grounding, which aims to accurately identify and localize objects in 3D space from 2D images and text queries. The VLM-Grounder operates without the need for additional training and leverages existing 2D foundation models. Initially, GPT-4V identifies images that contain the target object based on the provided text query. Subsequently, GPT-4V collaborates with Grounding-DINO to refine this selection and retrieve the target object. Finally, the identified object from multiple views is projected into 3D space, yielding the final 3D bounding box. This approach shows competitive performance on existing benchmarks.

**Summary Of Recommendation:**

The main concern comes from Table 1. I believe that all the performance scores should be calculated over the same data samples is a common sense. However, the results in Table 1 are obviously obtained from different experiment settings, which makes the conclusions untrustworthy. If the authors can provide experiment results under the same setting, I will raise my score.

---

### Author Rebuttal · Authors · 2024-08-09

We sincerely thank AC and all reviewers for their insightful feedback.

Our method has been recognized as "novel" (Reviewer oazi and Reviewer mxB2), "new" (Reviewer oE1Z), and "promising" (Reviewer oazi). We are gratified that the "powerful generalization capability" (Reviewer oazi), "reasonable pipeline" (Reviewer oazi), "well-articulated rationale behind the dynamic stitching and the visual-retrieval benchmark" (Reviewer mxB2), "informative experimental results" (Reviewer mxB2), and "easy-to-follow paper" (Reviewer oazi) have been acknowledged.

In the rebuttal PDF, we provide:
- Detailed error analyses and visualzations of the framework (Reviewer oazi and Reviewer mxB2)
- Matching visualization of PATS (Reviewer oazi)

---
**For Reviewer oazi and Reviewer mxB2**

**Q: Success rate and error analysis of the framework.**
**A:** We randomly sampled 50 samples from the ScanRefer evaluation data and manually inspected the results of each step across the framework. The success rates are shown in the table below.

| Step | Query Analysis | View Pre-Selection | Grounding | OV-Detection | Object Selection | SAM | Projection | Overall Acc@0.5 |
| :---- | :---- | :---- | :---- | :---- | :---- | :---- | :---- | :---- |
| Success Rate | 100% | 96% | 77% | 92% | 100% | 82% | 61% | 34% |

The standards for each step to be regarded as successful are illustrated as follows:
- Query Analysis: The analyzed query correctly identifies the target category and conditions.
- View Pre-Selection: The pre-selected views contain the target object.
- Grounding: The selected view contains the target object.
- OV-Detection: The detection results contain the correct detections of the target category.
- Object Selection: The target object is selected.
- SAM: The instance mask of the target object is neither over-segmented nor under-segmented.
- Projection: The IoU3D of the predicted bounding box and the GT bounding box is greater than or equal to 0.5.

We found that the main errors occur in the grounding, OV-detection, SAM, and projection modules. We provide the detailed error analyses and visualizations of these modules in the rebuttal PDF. We will incorporate these analyses into our revised paper accordingly.

---

We have addressed other individual comments in the following sections and hope our responses satisfy any concerns. Thank you once again for all your constructive insights\!

---

### Decision · Program_Chairs · 2024-09-04

**Decision:**

Accept

**Comment:**

All reviewers acknowledge the contributions of the presented work. The reviewers raised concerns regarding experiments showing results on a subset of 250 scenes from the validation set. The authors submitted a rebuttal where the baselines were evaluated on the same 250 validation examples. Overall, the performance of the proposed method is much higher than previous training-free methods on the benchmark. The authors are encouraged to only show the apples to apples comparisons on the 250 examples, else the experimental results on different validations sets are misleading. If those 250 examples are not randomly selected, they are also encouraged to run their method in the whole validation set. This will shed light on the limitations of the current prompts (extending the prompts  to cover the language variability of the dataset is a valid thing to add).